# The Enhanced Hydrogen Storage Capacity of Carbon Fibers: The Effect of Hollow Porous Structure and Surface Modification

**DOI:** 10.3390/nano11071830

**Published:** 2021-07-14

**Authors:** Sung-Ho Hwang, Young Kwang Kim, Hye-Jin Seo, Soon Moon Jeong, Jongwon Kim, Sang Kyoo Lim

**Affiliations:** 1Division of Energy Technology, DGIST, Daegu 42988, Korea; hsungho@dgist.ac.kr (S.-H.H.); kimyk1211@dgist.ac.kr (Y.K.K.); seohaejin511@dgist.ac.kr (H.-J.S.); smjeong@dgist.ac.kr (S.M.J.); 2Department of Fiber System Engineering, Yeungnam University, Gyeongsan 38541, Korea; 3Department of Interdisciplinary Engineering, DGIST, Daegu 42988, Korea

**Keywords:** hydrogen storage, microporosity, surface modification, carbon fiber

## Abstract

In this study, highly porous carbon fiber was prepared for hydrogen storage. Porous carbon fiber (PCF) and activated porous carbon fiber (APCF) were derived by carbonization and chemical activation after selectively removing polyvinyl alcohol from a bi-component fiber composed of polyvinyl alcohol and polyacrylonitrile (PAN). The chemical activation created more pores on the surface of the PCF, and consequently, highly porous APCF was obtained with an improved BET surface area (3058 m^2^ g^−1^) and micropore volume (1.18 cm^3^ g^−1^) compare to those of the carbon fiber, which was prepared by calcination of monocomponent PAN. APCF was revealed to be very efficient for hydrogen storage, its hydrogen capacity of 5.14 wt% at 77 K and 10 MPa. Such hydrogen storage capacity is much higher than that of activated carbon fibers reported previously. To further enhance hydrogen storage capacity, catalytic Pd nanoparticles were deposited on the surface of the APCF. The Pd-deposited APCF exhibits a high hydrogen storage capacity of 5.45 wt% at 77 K and 10 MPa. The results demonstrate the potential of Pd-deposited APCF for efficient hydrogen storage.

## 1. Introduction

Future energy security has been threatened by the decrease of fossil fuel resources and the polluting effects of their use. To resolve these issues, there have been tremendous efforts to invent effective renewable energy systems such as geothermal, wind, solar, biomass, and hydrogen energy. Among those candidates, hydrogen has attracted attention as it can replace fossil fuels by achieving both high energy effectiveness and eco-friendliness because of its high gravimetric energy density, and pure byproducts [1,2,3]. Despite these advantages, the application of hydrogen energy systems for practical use has key technical barriers to be overcome in storage efficiency, low volumetric storage density, durability [4,5,6,7], refueling time, cost, and safety issues [8,9,10,11]. To solve these problems, both selecting appropriate material and designing their structures, which are able to achieve high performance for hydrogen storage, are required.

In general, hydrogen is stored in one of four different ways: high-pressure compression method, liquefaction within a high-pressure tank, storage in a solid-state hydride, and storage in a porous material [12]. Although the compression and liquefaction were commonly used, the storage of hydrogen molecules via porous materials has been considered as an attractive technique due to the fast reaction kinetics, high adsorption capacity, and improved level of safety compared with that of the compressed-gas storage process. Moreover, liquefaction of hydrogen has its own challenges: high energy cost and risk of explosion, requiring the extremely low temperatures. In the case of chemical storage methods by metal hydrides, the efficiency in practical use is negatively affected by hysteresis phenomena between adsorption and desorption reaction, the high heat of reaction, and low thermal conductivity of metal hydrides [13]. Therefore, many porous materials, including polymer-derived carbons [13,14], zeolites [15], and metal-organic frameworks, have been extensively studied as materials for hydrogen storage [16]. Among these, carbon-based porous materials, especially porous carbon fibers, has drawn the attention of scientists as the most appropriate candidates for hydrogen storage because of their high pore volume, surface area, tunable texture structure, low gas-solid interaction, and excellent chemical and thermal stability [8,17]. Another important reason that carbon-based porous fibers are suitable as hydrogen storage materials is that it is easy to control their pore size with sub-nanometers favoring the adsorption of large numbers of hydrogen molecules. Basically, carbon-based porous materials with sub-nanometer pores can be prepared by the successive process of carbonization and chemical activation of polymer-containing solutions. For more effective hydrogen storage, various preparation strategies, including template-based methods, spray pyrolysis [18], and rapid thermal carbonization [19] have also been introduced as a means of controlling the pore size of carbon materials with a range of sub-hundred nanometers. Another strategy for enhancing the hydrogen storage capacity of carbon materials is the deposition of Pd nanoparticles (which have a strong affinity for hydrogen) on the surface. The improvement of the hydrogen storage capacity is closely related to the spillover effect [20] defined as the transfer of dissociative hydrogen atoms from the surface of a metal to that of the carbon substrate by a hydrogen concentration gradient induced by hydrogen affinity [21].

In this study, we developed a new strategy to achieve a high-capacity means of hydrogen storage based on carbon fibers by controlling the surface porosity (Figure 1). The core strategy is associated with forming pores both exterior and interior of the carbon fiber. First, we prepared polyvinyl alcohol/polyacrylonitrile (PVA/PAN) bi-component fibers via wet spinning, after which we removed the component (PVA) to create mainly internal pores of the fibers by immersing them in water, and by drying and calcinating them. The difference in the solubility of PVA and PAN in water is closely related to the formation of the internal pores. Subsequently, we carried out further chemical activation to create micropores and mesopores both interior and exterior of the carbon fiber. Finally, the Pd nanoparticles were electrodeposited on the surface of the carbon fiber for further improvement of the hydrogen storage capacity. Herein, we discuss the effect of the modified morphologies of the samples by successive processes—PVA removal, calcination, chemical activation and deposition of Pd nanoparticles—on their storage capacity in detail.

## 2. Materials and Methods

### 2.1. Materials

Dimethyl sulfoxide (DMSO, Duksan, Seoul, Korea, 99%), methyl alcohol (MeOH, Duksan, Seoul, Korea, 99%), potassium hydroxide (KOH, Duksan, Seoul, Korea, 99%), palladium chloride (PdCl_2_, Aldrich, St. Louis, MO, USA, 99%), hydrochloric acid (HCl, Daejung, Siheung, Korea, 0.1 N), polyacrylonitrile (PAN, Jilin Carbon Co., Ltd. Dalian, China), low-molecular-weight polyvinyl alcohol (PVA_L, Aldrich, St. Louis, MO, USA, 99%, molar weight (M_w_); 85,000–124,000, degree of polymerization (D_P_); 2000, degree of hydrolysis (D_H_); 99%) and high-molecular-weight polyvinyl alcohol (PVA_H, Aldrich, St. Louis, MO, USA, 99%, M_w_; 146,000–186,000, D_P_; 4000, D_H_; 99%) were used in this study.

### 2.2. Preparation of PVA/PAN Bi-Component Fiber and Porous PAN Fiber (PPF)

The preparation of carbon fibers is summarized into four steps: (i) production of bicomponent polymeric solutions with different solubility in water; (ii) formation of PVA/PAN bi-component polymeric fibers via wet spinning (Dissol, Korea); (iii) removal of on polymeric component to create pores within the fibers; and (iv) calcination of porous polymeric fibers. Polymeric solutions of PVA_H (or PVA_L, 15 wt%) and PAN (15 wt%) were prepared by dissolving them in DMSO at 50 °C. Then, dope solutions (thermodynamically stable polymer solution formed by a polymer and solvent mixture) [22] were prepared for wet spinning by mixing the PAN and PVA_H (or PVA_L) bi-component solutions at a weight ratio of 5:5, 7:3, 9:1 and 1:0 (for preparing a non-porous PAN fiber as a reference sample). The dope solutions were then extruded through a spinneret with 32 orifices (diameter; 0.1 mm) and immersed in a MeOH coagulant at 25 °C. Subsequently, the resultants were drawn out through three continuous rollers at drawing speeds of 1, 3, and 3.4 m min^−1^ to form PVA/PAN bi-component fibers Figure 1a, which were then dried at 90 °C and wound onto bobbins. To fabricate the porous PAN fiber (PPF) Figure 1b, the prepared bi-component fibers were immersed in distilled water at 100 °C for 5 h to selectively remove PVA_H (or PVA_L) from the fibers and were finally dried in an oven at 50 °C for 24 h.

### 2.3. Preparation of Porous Carbon Fiber (PCF) and Activated Porous Carbon Fiber (APCF)

We considered that the molecular weight of PVA and the ratio between PVA and PAN are important factors with regards to optimizing the carbon fiber porosity. With this in mind, porous carbon fibers (PCFs) were prepared by the carbonization of the as-prepared PPFs in a silicon carbide tube furnace. For the carbonization of the PPFs, the temperature ramped with a constant rate of 5 °C min^−1^ from 25 °C to 280 °C under airflow and stabilized at 280 °C for 1 h. After that, the temperature was raised from 280 °C to 1200 °C with the temperature ramping rate of 5 °C min^−1^ under nitrogen gas and remained at the 1200 °C for 1 h and cooled to 25 °C. The prepared carbon fibers were denoted PCF_H 0.5, PCF_H 0.3, PCF_H 0.1 (or PCF_L 0.5, PCF_L 0.3, PCF_L 0.1), and CF depending on the concentration of PAN and PVA_H (or PVA_L) in the dope solutions, with weight ratios of 5:5, 7:3, 9:1 and 1:0 (PAN alone), respectively in Figure 1c.

For the creation of more pores, chemical activation of the PCF was carried out using a KOH solution. A bundle of PCFs (0.5 cm × 5 cm, 0.2 g) was immersed in a KOH solution (8 M, 50 mL) at 60 °C for 4 h and then dried at 100 °C for 24 h. Then, the samples were calcined in a tube furnace under nitrogen flow. The temperature in the furnace was raised at a constant rate of 5 °C min^−1^ from 25 °C to 900 °C and remained there for 1 h. To remove any residual potassium from the activated PCFs (APCFs), the fibers were immersed in an HCl (0.1 N, 50 mL) solution for 30 min and then washed repeatedly with deionized water until the pH of the samples became neutral. Finally, the APCFs were obtained after the drying of the samples at 100 °C for 12 h in an oven. The as-prepared APCFs were denoted APCF_H 0.5, APCF_H 0.3, APCF_H 0.1 (or APCF_L 0.5, APCF_L 0.3, APCF_L 0.1), ACF depending on the concentration of PAN and PVA_H (or PVA_L) in the dope solution with weight ratios of 5:5, 7:3, 9:1 and 1:0 (PAN alone), respectively (Figure 1d).

### 2.4. Electrodeposition of Pd Nanoparticles on APCF

Among the APCFs, APCF_H 0.3 possessed the highest surface area. Therefore, Pd nanoparticles were electrodeposited on APCF_H 0.3 as reported in the literature to investigate the effect of Pd nanoparticles on the hydrogen storage capacity [23]. After placing a working electrode, bundles of APCF_H 0.3 (0.5 cm × 5 cm, 0.2 g), and Pt wire counter electrode into aqueous KCl (0.1 mM, 0.5 mL) solutions containing different concentrations of PdCl_2_ (0.1, 0.5, and 1 mM), the electric field was applied at−1.0 V versus the reference electrode (Ag/AgCl) for 30 min using a potentiostat (VSP, Bio-logic Science Instrument, France). The Pd-deposited APCF was obtained after washing with distilled water and then drying in the air (Figure 1e). Hereafter, the samples are denoted Pd 0.1/APCF_H 0.3, Pd 0.5/APCF_H 0.3 and Pd 1/APCF_H 0.3 depending on the concentration of the PdCl_2_ solution (0.1, 0.5, 1 mM, respectively).

### 2.5. Characterizations

The surface morphology and elemental substances of the samples were analyzed by scanning electron microscopy (SU-8020, Hitachi, Tokyo, Japan) and transmission electron microscopy (HF-3300, Hitachi, Tokyo, Japan). The nitrogen adsorption-desorption isotherms at 77 K were used to analyze the pore size distribution, pore-volume, and specific surface area of the samples (ASAP 2020, Micromeritics, Norcross, GA, USA). The pore size distribution and micropore volume were estimated by the Horvath–Kawazoe method [24]. The hydrogen storage capacity of the samples was measured by the gravimetric method at 77 K (ISOSORP-HyGrA, Waters Corporation, Milford, MA, USA). Correlation between the hydrogen storage capacity and the specific area of the as-prepared carbon samples is calculated as the following equation:Hydrogen storage capacity (wt%) = K × SSA _carbon fiber_ (m^2^ g^−1^) (1)
where K is a slope between hydrogen storage capacity and specific surface are of the as-prepared carbon samples and SSA _carbon fiber_ is the specific surface area of the as-prepared carbon samples.

## 3. Results and Discussion

### 3.1. Surface Characterization

For the successful fabrication of highly porous carbon fibers for hydrogen storage, the optimized PPFs should be first fabricated by selectively removing the PVA from PVA/PAN bi-component fibers. From this point of view, firstly, the effect of the molecular weight of PVA was investigated. From the cross-sectional views of the PPFs of Figure 2a,b, it could be seen that channel-like pores were formed on the PVA/PAN bi-component fiber via the selective removal of the PVA component. It was notable that the larger cracks were observed in PPFs prepared with a larger amount and lower molecular weight of PVA, which might be due to the aggregation of PVA. On the other hand, the morphologies of PCFs prepared by calcining these PPFs were shown in Figure 2c,d. All the PCFs exhibited channel-like macropores, showing the larger diameter in PCFs prepared with the lower molecular weight of PVA than those prepared with a higher molecular weight of PVA, similarly as shown in PPFs. It was reported that the average diameter of aggregated PVA increased as the molecular weight and viscosity decreased [25,26]. The surface characteristics according to the amount and molecular weight of PVA are listed in Table 1. As expected in Figure 2, it is confirmed in Table 1 that PCF_H has a larger surface area and porosity than PCF_L. In Table 1, it is also seen that PCF prepared with a PAN/PVA ratio of 7:3 and showed the best surface area and pore volume. It is natural that the surface area would become larger as PVA was the more used, since the number of pores, which resulted from the removal of PVA in PPF, might be increased. But the aggregation of PVA might also make the size of the pores larger rather than increase the number of them, leading to a decrease in surface area. As described above, the aggregation of PVA became increased in the largest amount and lower molecular weight of PVA. As a result, PCF_H 0.3 showed the best surface area due to such an effect and was expected to show the best hydrogen storage capacity. Actually, it was confirmed in Appendix A that the hydrogen storage capacity was closely related to surface area and pore volume. Therefore, the effect of various treatments such as chemical activation or metal deposition will be discussed with the sample obtained by chemical activation and metal deposition treatment on PCF_H 0.3.

To improve the porosity further, PCF_H 0.3 was chemically activated with KOH treatment. SEM images of various carbon fibers were presented in Figure 3. The PCF shows prominently porous morphology due to the removal of PVA compared with CF (prepared by carbonization of mono-component PAN). In addition, after the treatment of chemical activation in PCF, the smooth surface in PCF was confirmed to be transformed into a highly porous surface in APCF (Figure 3f,i), which would lead to a higher surface area and porosity. The mechanism of the activation process of PCF via KOH is generally understood by the following three-step explanation reported in the literature elsewhere [27,28,29]. (i) Formation of the pore network in carbon by the etching process of a carbon framework via the redox reaction between carbon and potassium compounds; (ii) additional formation of pores by gasification of carbon promoted by the reaction between carbon and generated gases (CO_2_ and H_2_O); (iii) irreversible expansion of a carbon lattice by intercalation of metallic potassium into the carbon lattice. The surface area and microporosity in various types of carbon fiber, with and without chemical activation and Pd deposition, are summarized in Table 2.

PCF_H 0.3 exhibits a high BET surface area of 889 m^2^ g^−1^ and a pore volume of 0.47 cm^3^ g^−1^. The surface area is almost 80-fold that of the non-porous CF (11 m^2^ g^−1^). Importantly, the molecular weight of the PVA used in the dope solution also influences the microporosity of the sample. The fraction of micropores relative to the total pore volume increases to around 13.1% (Table 1) [34].

Alkali-mediated activation induces a significant improvement in both the BET surface area and microporosity. APCF_H 0.3 exhibits an ultra-high BET surface area of 3058 m^2^ g^−1^ and a pore volume of 1.55 cm^3^ g^−1^ (Figure 4).

These values are improved over those of PCF_H 0.3 by factors 3.4 and 3.3, respectively, due to the formation of micropores and mesopores on the porous carbon fibers. Comparing APCF_H0.3 and ACF, macropore creation prior to chemical activation improves the BET surface area and pore volume by factors of 9.7 and 9.8, respectively. The micropore size distribution of the samples is shown in Figure 4b. The pore sizes of PCF_H 0.3 are centered at about 1.08 nm, making them ideal for hydrogen storage. Moreover, the number of micropores in APCF_H 0.3 with a size of 1.04 nm is greatly increased. This indicates that alkali-mediated activation plays a crucial role in reducing the pore size of the carbon fiber and increasing the micropore fraction. The above results show that carbon fibers with appropriate surface area and microporosity for improved hydrogen storage capacity can be successfully achieved by creating various pore sizes via (i) PVA removal from bi-component fiber and (ii) chemical activation of the porous carbon fiber.

### 3.2. The Hydrogen Storage Capacity of PCF, APCF, and Pd Deposited APCF

In the previous section, we examined the effects of selective removal of PVA from bi-component fiber (PVA/PAN fiber), carbonization of PPFs and alkali activation of PCFs on their surface area and microporosity. Now, we discuss the relationship between these properties (surface area and microporosity) and the hydrogen storage capacity of the samples and then compare the hydrogen storage capacity of activated carbon fibers, reported elsewhere, with our results. The hydrogen storage behavior of the samples at 77 K and 10 MPa is given in Table 2 and Figure 4c and Appendix A. The hydrogen storage behavior exhibits a rapid increase in the capacity below 500 kPa, but the value becomes saturated at 2 MPa. In Figure 4c, CF, ACF, PCF_H 0.3, and APCF_H 0.3, with surface areas of 11, 316, 889, and 3058 m^2^ g^−1^ respectively, have hydrogen storage capacities of 0.07, 1.35, 2.87, and 5.14 wt%, respectively, at 77 K and 10 MPa. The results reveal that the hydrogen storage capacity is proportional to the BET surface area, which is also supported by Chahine’s rule that it can adsorb 2.00 × 10^−3^ wt% hydrogen gas per unit specific surface area of carbon materials [17,35]. The previous theoretical studies upon hydrogen storage reveal that carbon materials can adsorb hydrogen up to 2.28 × 10^−3^ wt% for every unit surface area, which is the limit of the amount of hydrogen by monolayer adsorption [36]. In a similar manner as above, we can estimate the hydrogen adsorption capacity of the unit-specific surface area of carbon fibers. Equation (1) showed that as-prepared carbon fibers can adsorb the 1.41 × 10^−3^ wt% hydrogen gas in every unit specific surface area at 77 K (Appendix A).

We especially observed that APCF_H 0.3 exhibits the best hydrogen storage capacity with the exception of Pd-deposited samples. Such a highly improved hydrogen storage capacity might originate from microporosity, which could play a crucial role in hydrogen storage. It was expected that micropores with a sufficient quantity of hydrogen storage existed in APCF_H 0.3 relative to other samples. Actually, it was observed that the chemical activation treatment of PCF_H 0.3 forms a narrower micropore with an average micropore width of 1.05 nm in APCF_H 0.3, which expands the micropore volume by a factor of 2.1 (Table 2). A number of micropores with a range of 0.8–1.2 nm in APCF_H 0.3 will be able to interact strongly with the hydrogen molecules via Van der Waals interactions, leading to an improvement in the enthalpy of adsorption [13,37]. It has been reported that hydrogen storage in porous carbon materials via physisorption is dominated by pores in the sub-nanometer range [38,39]. Especially, it is reported that narrow pores under 1.5 nm improve the hydrogen storage because the narrow pores enhance the interaction energy between hydrogen molecules and the pore walls of carbon materials [40]. However, the pore size below 0.5 nm is not sufficient for hydrogen storage because dynamic hydrogen molecules would not enter the inside of the pore [41]. Therefore, several groups have conducted experiments to identify the optimal pore diameter for hydrogen storage and conclude that it is 0.5–0.71 nm [42,43,44,45]. Moreover, theoretically, it has been estimated that the optimal pores in carbon materials for hydrogen storage should have a width of at least 0.56 nm at 77 K, regardless of the pressures [46]. Several studies have also proven that the hydrogen storage capacity of carbon fibers is proportional to the micropore volume of the carbon materials [47,48]. As shown in Appendix Ab, the hydrogen storage capacity of the samples shows a linear dependence on the micropore volumes of the samples. Through these observations, it was found that the largest micropore volume of APCF_H 0.3 enhances the hydrogen storage capacity (Table 2). Therefore, we can conclude that both the micropore size, especially in the case of a 1-nm pore width, as well as the micropore volume, play a crucial role in increasing the hydrogen storage capacity.

For further improvement of hydrogen capacity, Pd nanoparticles were electrodeposited onto the APCF_H 0.3. When Pd nanoparticles are deposited on APCF_H 0.3, the BET surface area falls to 2611 m^2^ g^−1^, the total pore volume falls to 1.42 cm^3^ g^−1^, and the percentage of micropore volume in total pore volume falls by 4.1%. In addition, the number of pores that would constitute effective hydrogen storage (0.8–1.2 nm) is slightly decreased. A decreasing tendency of the surface area, pore-volume, and fraction of micropores are also observed in the Pd-deposited samples as the concentration of PdCl_2_ is increased. This indicates that the Pd nanoparticles cover the pores of APCF_H 0.3 (Table 3) [49]. As expected, the Pd nanoparticles were really found as aggregates on the APCF_H 0.3 and to partially cover most of the pores on the surface of the fibers as the concentration of the PdCl_2_ solution increases (Appendix A and Appendix A). The average size of Pd nanoparticles in the representative sample, Pd 0.5/APCF_H 0.3, was 41.7 nm. The nanoparticles were mainly composed of Pd (111) and Pd (200) planes corresponding to a lattice spacing of 0.19 and 0.22 nm, respectively (Figure 5) [50]. The hydrogen storage capacity is increased after the electrodeposition of Pd nanoparticles on the surface of the APCF_H 0.3, at a ratio of up to 5.45 wt% (for a PdCl_2_ concentration of 0.5 nm, Appendix A). Interestingly, when the PdCl_2_ concentration is increased, the hydrogen storage capacity saturates. It seemed that electro-deposited Pd nanoparticles partially blocked the micropores on the APCF_H 0.3, resulting in a decrease of the fraction of micropore volume that could be a helpful factor for hydrogen storage capacity. Simultaneously, the hydrogen storage capacity increases with the spillover effect of Pd nanoparticles that involves the following steps: (i) chemisorptive dissociation of hydrogen molecules on the Pd nanoparticles; (ii) diffusion of hydrogen atoms from Pd nanoparticles (hydrogen-rich) to the carbon substrate (hydrogen-poor); and (iii) storage of hydrogen atoms at pores of carbon substrate [20,21,51]. Therefore, between two conflicting effects, it is necessary to investigate which effect dominates the hydrogen storage capacity. To estimate the extent of the improvement in the hydrogen storage capacity caused by the spillover effect alone, the hydrogen storage capacity of Pd 0.5/APCF_H 0.3 via physisorption, 4.78 wt%, was calculated by the above-mentioned empirical equation (Equation (1), Appendix A). The assumed improvement in hydrogen storage capacity resulting from the spillover effect was calculated by subtracting the calculated hydrogen storage capacity from the experimental result. Hence, the improvement caused by the spillover effect was determined to be around 0.67 wt%. These results indicate that there is a competitive relationship between the physisorption by alkali activation and chemisorption by the spillover effect [52,53,54]. Therefore, for the application of metal-deposited carbon fiber in hydrogen storage, the effect of metal nanoparticles on micropore volume and pore size of the metal deposited-carbon fiber would have to be considered together with a spillover effect [55,56].

### 3.3. APCF with an Excellent Hydrogen Storage Capacity

As shown in Figure 4d, the hydrogen storage capacity and surface area of carbon fibers reported elsewhere were compared with those of our samples [30,31,32,33]. The literature states that both carbon fibers and activated carbon fibers have small surface areas (150 and 265 m^2^ g^−1^, respectively) and hydrogen storage capacity (0.4 and 0.6 wt%, respectively) due to their intrinsic morphologies. However, here, the optimal porous carbon fiber (PCF_H 0.3) and activated porous carbon fiber (APCF_H 0.3) have an extremely larger surface area and higher hydrogen storage capacity than carbon fibers and activated carbon fibers (Table 2). Interestingly, the hydrogen storage capacity of APCF_H 0.3 is similar to that of an activated carbon aerogel produced using CO_2_, 5.3 wt% [17]. Furthermore, the hydrogen storage capacity of APCF_H 0.3, 5.14 wt%, is higher than that of carbon fibers reported elsewhere [30,31,32,33]. This comparison indicates that our strategy for controlling the porosity of carbon fibers is extremely promising for advancing hydrogen storage.

## 4. Conclusions

In the present study, porous carbon fiber and activated porous carbon fiber were prepared via the carbonization and alkali activation of PVA/PAN bi-component fibers from which the PVA had been selectively removed. We found that the pore size of the carbon fiber decreased as the molar weight of the PVA increased. The surface activation of porous carbon fiber leads to a tremendous increase in the surface area (3058 m^2^ g^−1^) and micropore volume (1.18 cm^3^ g^−1^) of the fiber. The improved surface area and microporosity of the carbon fiber enhanced the hydrogen storage capacity via physisorption to 5.14 wt%. To the best of our knowledge, this value is higher than those of previously reported carbon fibers. For further improvement of hydrogen storage capacity, Pd nanoparticles were electrodeposited on the chemically activated porous carbon fiber. It was revealed that the hydrogen storage capacity of Pd-deposited activated porous carbon fiber is 5.45 wt% at 77 K and 10 MPa. In summary, we developed a new strategy to achieve a high-capacity means of hydrogen storage on the basis of carbon fibers by controlling the surface porosity. The core strategy is associated with forming pores both exterior and interior of the carbon fiber using the polyvinyl alcohol/polyacrylonitrile (PVA/PAN) bi-component fibers. The difference in the solubility of PVA and PAN in water is closely related to the formation of the internal pores. This method is found to be very effective to improve the pore characteristics of carbon fibers by way of successive chemical activation. Finally, the Pd nanoparticles were electrodeposited on the surface of the carbon fiber for further improvement of the hydrogen storage capacity; however, it was also found that the enhancement in hydrogen storage capacity via a spillover effect due to deposition of Pd nanoparticles is relatively low (~0.3 wt%) due to deposition of Pd nanoparticles, since the micropores of the activated porous area were partially covered by the deposition of Pd nanoparticles. For the application of metal-deposited carbon fiber in hydrogen storage, the effect of metal nanoparticles on pore characteristics in metal deposited-carbon fiber would have to be considered together with spillover effect. We believe that the proposed method which involves the control of the surface morphology of the carbon fibers and Pd deposition on their surfaces could provide high hydrogen storage capacity.

## Figures and Tables

**Figure 1 nanomaterials-11-01830-f001:**
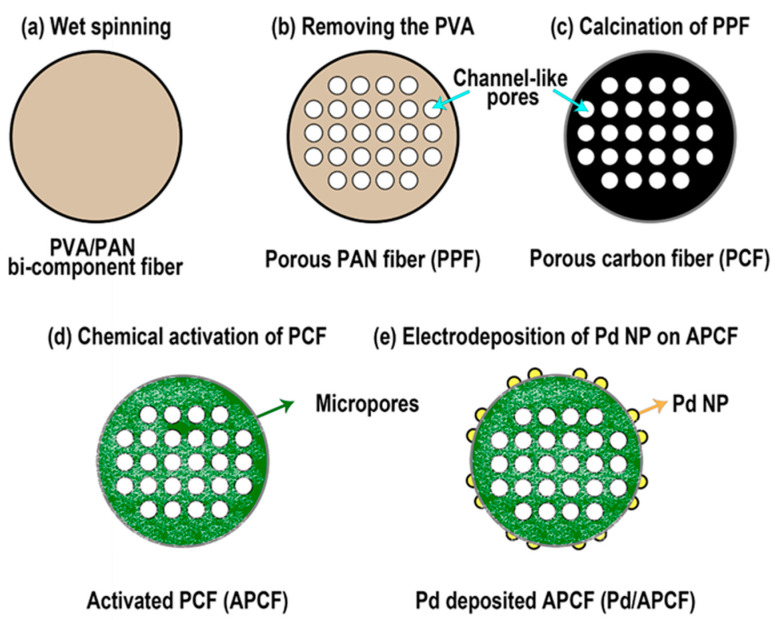
Cross-sectional schematic illustration of (**a**) PVA/PAN bi-component fiber; (**b**) Porous PAN fiber (PPF) fabricated by removing the PVA component from the PVA/PAN bi-component fiber; (**c**) Porous carbon fiber (PCF) prepared by calcination of the PPF; (**d**) Activated PCF (APCF) prepared by chemical activation of PCF; (**e**) Palladium electrodeposited APCF (Pd/APCF).

**Figure 2 nanomaterials-11-01830-f002:**
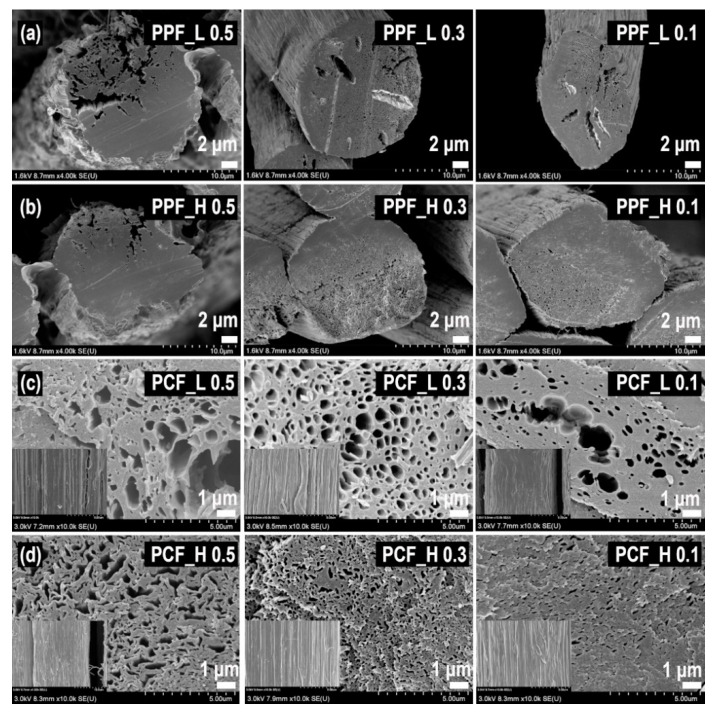
Cross-sectional SEM images of (**a**,**b**) PPFs (PVA-removed PAN/PVA bi-component fibers) and (**c**,**d**) porous carbon fibers (PCFs; carbonized PPFs), in which (**a**,**c**) low molecular weight PVA and (**b**,**d**) high molecular weight PVA were used when preparing the bi-component fibers. The bi-component fibers were prepared with PAN/PVA weight ratio of 5:5, 7:3, and 9:1 from left to right. The insets in (**c**,**d**) show side-view SEM images of each PCF.

**Figure 3 nanomaterials-11-01830-f003:**
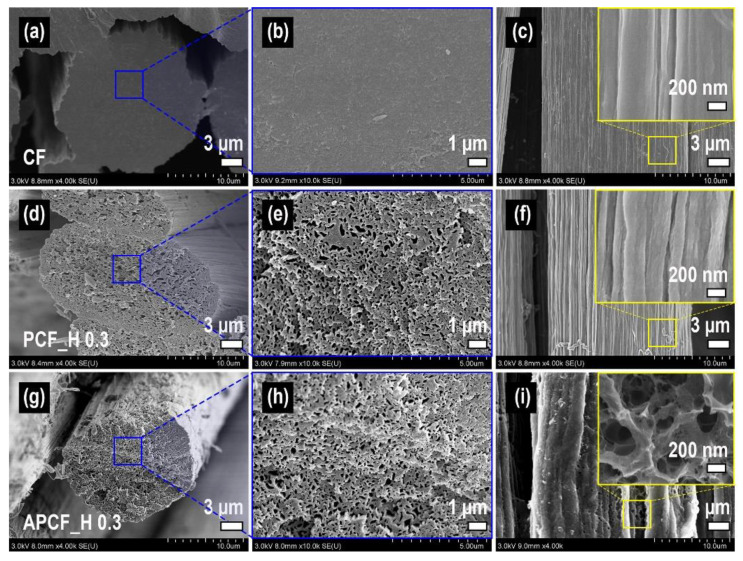
FE-SEM images of (**a**–**c**) carbon fiber prepared by carbonization of mono-component PAN, (**d**–**f**) PCF_H 0.3, and (**g**–**i**) APCF_H 0.3. (**a**,**d**,**g**) cross-section, (**b**,**e**,**h**) enlarged cross-section, (**c**,**f**,**i**) side view (inset: enlarged side view).

**Figure 4 nanomaterials-11-01830-f004:**
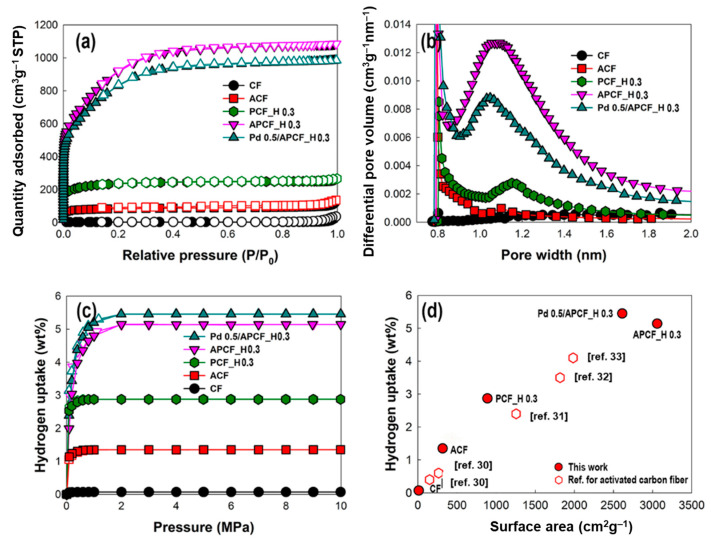
(**a**) N_2_ adsorption–desorption isotherm (empty–filled markers); (**b**) pore size distributions and (**c**) hydrogen uptakes of CF, ACF, PCF_H 0.3, APCF_H 0.3, and Pd 0.5/APCF_H 0.3 (PdCl_2_ solution concentration: 0.5 mM); (**d**) Comparison of hydrogen uptake capacity (measured at 77 K and 10 MPa) vs. surface area of activated carbon fiber (empty hexagons), as reported in the literature, and as determined in the present study (filled red circles).

**Figure 5 nanomaterials-11-01830-f005:**
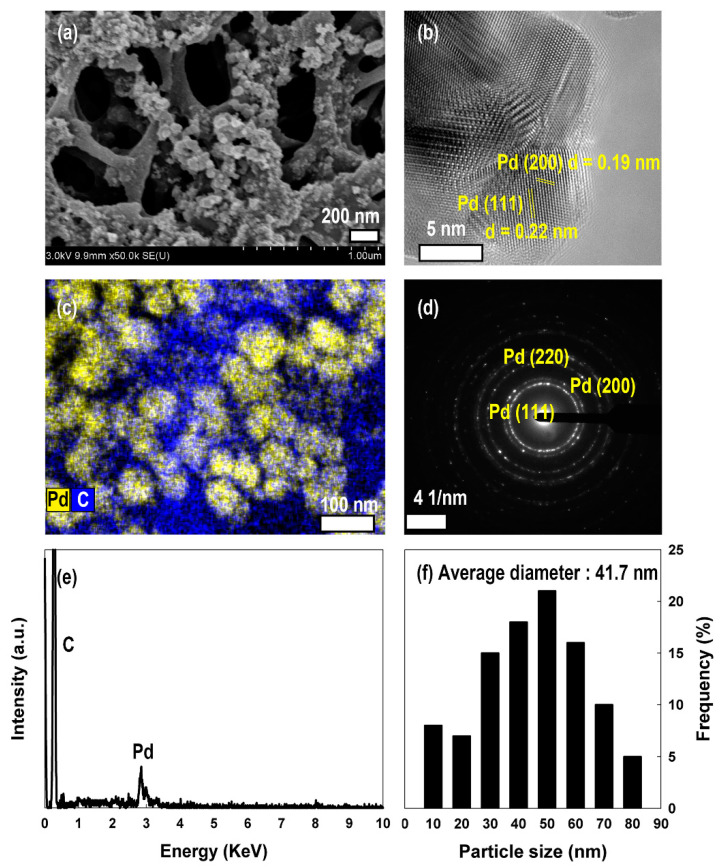
(**a**) SEM image, (**b**) TEM image, (**c**) EDS elemental mapping image, (**d**) SAED pattern, and (**e**) EDS spectrum of Pd 0.5/APCF_H 0.3. (**f**) Diameter distribution of Pd nanoparticles in the sample, Pd0.5/APCF_H 0.3.

**Table 1 nanomaterials-11-01830-t001:** Surface area and microporosity of porous carbon fibers (PCF) prepared from dope solutions with different weight ratios of PAN and PVA.

Sample	Dope Solution	Weight Ratio	S_BET_ ^1^ (m^2^ g^−1^)	V_p_ ^2^ (cm^3^ g^−1^)	V_micro_ ^3^ (cm^3^ g^−1^)
PCF_L 0.5	PAN/PVA_L	5:5	201	0.25	0.12 (48.0)
PCF_L 0.3	PAN/PVA_L	7:3	424	0.27	0.16 (59.2)
PCF_L 0.1	PAN/PVA_L	9:1	110	0.06	0.02 (33.3)
PCF_H 0.5	PAN/PVA_H	5:5	377	0.27	0.14 (51.9)
PCF_H 0.3	PAN/PVA_H	7:3	889	0.47	0.34 (72.3)
PCF_H 0.1	PAN/PVA_H	9:1	798	0.46	0.25 (54.3)

^1^ S_BET_: BET specific surface area, ^2^ V_P_: Pore volume, estimated as P/P_0_ ≈ 0.99. BET specific surface area, ^3^ V_micro_: Micropore volume, determined by the Horvath–Kawazoe (HK) method; the percentage of the total pore volume constituted by micropores is given in parentheses.

**Table 2 nanomaterials-11-01830-t002:** Surface area, microporosity, and hydrogen storage capacity of samples at 77 K and 10 MPa in comparison with reported literature.

Samples	S_BET_ ^1^ (m^2^ g^−1^)	V_p_ ^2^ (cm^3^ g^−1^)	V_micro_ ^3^ (cm^3^ g^−1^)	W_average_ ^4^ (nm)	H_2_ Uptake (wt%)	Reference
CF	11	0.06	0.01 (16.7)	1.81	0.07	This work
ACF	316	0.21	0.12 (57.1)	1.13	1.35	This work
PCF_H 0.3	889	0.47	0.34 (72.3)	1.12	2.87	This work
APCF_H 0.3	3058	1.55	1.18 (76.1)	1.05	5.14	This work
Pd 0.5/ APCF_H 0.3	2611	1.43	1.03 (72.0)	1.03	5.45	This work
NF	150	-	-	-	0.4	[30]
ANF	265	-	-	-	0.6	[30]
ARCF1	1256	-	-	-	2.4	[31]
ACF A20	1817	-	-	-	3.5	[32]
ACF A20	1984	-	-	-	4.1	[33]

^1^ S_BET_: BET specific surface area, ^2^ V_P_: Pore volume, estimated as P/P_0_ ≈ 0.99. BET specific surface area, ^3^ V_micro_: Micropore volume, determined by the Horvath–Kawazoe (HK) method; the percentage of the total pore volume constituted by micropores is given in parentheses, ^4^ W_average_: Average micropore width, determined using the Saito Foley method.

**Table 3 nanomaterials-11-01830-t003:** Surface area, microporosity, and hydrogen storage capacity at 77 K and 10 MPa of Pd-deposited APCF_H 0.3.

Sample	[PdCl_2_] (mM)	S_BET_ ^1^ (m^2^ g^−1^)	V_p_ ^2^ (cm^3^ g^−1^)	V_micro_ ^3^ (cm^3^ g^−1^)	H_2_ Storage (wt%)
APCF_H 0.3	0	3058	1.55	1.18 (76.1)	5.14
Pd 0.1/APCF_H 0.3	0.1	2760	1.44	1.05 (72.9)	5.30
Pd 0.5/APCF_H 0.3	0.5	2611	1.43	1.03 (72.0)	5.45
Pd 1/APCF_H 0.3	1	2503	1.40	1.00 (71.4)	5.17

^1^ S_BET_: BET specific surface area, ^2^ V_P_: Pore volume, estimated as P/P_0_ ≈ 0.99. BET specific surface area, ^3^ V_micro_: Micropore volume, determined by the Horvath–Kawazoe (HK) method; the percentage of the total pore volume constituted by micropores is given in parentheses.

## Data Availability

Not applicable.

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
