# Peer review of "The Enhanced Hydrogen Storage Capacity of Carbon Fibers: The Effect of Hollow Porous Structure and Surface Modification"

_nanomaterials, 2021, doi:10.3390/nano11071830_

Round 1

Reviewer 1 Report

In this article, Hwang et al. describe the fabrication of porous carbon fibers and their hydrogen storage capacity. The approach is interesting and original, and the results (in terms of hydrogen storage capacity) are excellent.

However, some sections of the paper need serious re-editing and proof reading as the grammatical and typographical mistakes make it difficult to understand. In particular, the introduction needs to be carefully checked and rewritten where appropriate (the other sections of the article are much better written for some reason). For example, on line 27: what does "Carbon Future energy security" mean? The sentence in lines 30-32 is not clear, and "ecofriend" and "lightweight" are not used appropriately. Etc. 

In the keywords: "sensor" is never mentioned and should be removed; surface "mordification" should be replaced by "modification".

More generally, and beyond these comments on grammar/typography, I would ask the authors to address the following comments before considering the article for publication:

  1. Introduction: the use of carbon fibers is interesting, however not frequently reported for hydrogen storage. Could the authors explain the rationale for using/creating carbon fibers, their advantages and disadvantages, etc?
    Line 32: hydrogen is indeed lightweight, which is a challenge more than an advantage once we consider the volumetric storage density. Please comment on this.
    Line 37: in addition to liquefaction, compression should be indicated as a typical hydrogen storage technique. Liquafaction is also used, but requires much lower temperatures.
    Figure 1: in the bottom right panel, Pd nanoparticles appear deposited at the surface of the fibers rather than inside the pores; could the authors comment in the text, where appropriate, whether this is the case or if Pd NPs also enter the pores?

  2. Materials and Methods. In line 134 (and elsewhere in the text) the authors refer to "best porosity" which is not clear as it could refer to surface area, pore size distribution, pore volume etc. Please use a more specific description - I assume this refer to "highest surface area" in this context.
  3. Results and discussion. Line 245: unit is missing after 1.08; 
    Regarding the method: could the authors discuss the possibility of doing alkaline activation+carbonisation in a single step (steps c and d on Figure 1)? Is it something worth trying, that has been attempted, etc?
    Line 285: a reference is needed after the statement "it is reported that narrow pores ..."
    Line 287: the sentence starting with "However, the heat of adsorption..." is not clear: small pores (<1 nm) are generally associated to a higher heat of adsorption (or to stronger interactions between the gas and the adsorbent) which is generally seen as a good thing for adsorption-based storage. Please clarify this sentence.
    Line 316: The sentence should read "pore volume falls by 4.1%" instead of "to 4.1%" based on numbers from Table 2.
    In the section about palladium deposition, the authors should comment on whether the addition of palladium is worth the cost/effort considering the relatively small increase in hydrogen storage capacity (this would also fit in the conclusion section, see my comment below).
  4. Conclusions. This section is mostly a summary of the rest of the article. The authors should give here a broader perspective about their findings in the general context of porous materials for hydrogen storage. For example, where would it be advantageous to use such fiber-based materials? Is it worth adding palladium? Critically commenting the results would significantly enrich the conclusion and the article itself.
  5. In general: most figures are low-resolution and sometimes difficult to interpret; can the authors supply high-resolution version of the figures for the published version of the article. 

Overall, this article is interesting and deserves publication in Nanomaterials. I recommend it to be "reconsidered after major revisions" owing to the large number of modifications that I am requesting, even if most of them are relatively minor. 

Author Response

Dear Reviewers,

First of all, I’d like to express my sincere and thanks to Reviewers who read my manuscript with deep concerns and to valuable comments.

According to reviewer’s comments, the manuscript was revised and checked to avoid some typical mistakes and expression or notation to cause misunderstanding.

We appreciate kind comments and suggestion of reviewer again. Our responses to these comments are listed below, and the manuscript has been accordingly revised. All the changes were represented in blue color in revised manuscript.

Thank you for your kindness and help

Sincerely yours,

Response to the Reviewer 1’s comments

In this article, Hwang et al. describe the fabrication of porous carbon fibers and their hydrogen storage capacity. The approach is interesting and original, and the results (in terms of hydrogen storage capacity) are excellent.

However, some sections of the paper need serious re-editing and proof reading as the grammatical and typographical mistakes make it difficult to understand. In particular, the introduction needs to be carefully checked and rewritten where appropriate (the other sections of the article are much better written for some reason). For example, on line 27: what does “Carbon Future energy security” mean? The sentence in lines 30-32 is not clear, and “ecofriend” and “lightweight” are not used appropriately. Etc. 

In the keywords: “sensor” is never mentioned and should be removed; surface “mordification” should be replaced by “modification”.

More generally, and beyond these comments on grammar/typography, I would ask the authors to address the following comments before considering the article for publication:

Response Thank you for the valuable advice on the manuscript. As the reviewer pointed out, the grammatical errors in our manuscript were corrected. In special, the introduction was checked and rewritten. And, the keywords were also removed or changed according to reviewer’s comments.

Page 1, line 24

Keywords: Hydrogen storage; microporosity; surface modification; carbon fiber.

Page 1, line 27

Future energy security has~

Page 1, line 31

~ Among those candidates, hydrogen has attracted attention as it can replace fossil fuels by achieving both high energy effectiveness and eco-friendliness because of its high gravimetric energy density, and pure byproducts [1-3]. Despite these advantages, the application of hydrogen energy systems for practical use has key technical barriers to be overcome in storage efficiency, low volumetric storage density, durability[4-7], refueling time, cost, and safety issues [8-11].~

Comment 1. Introduction: the use of carbon fibers is interesting, however not frequently reported for hydrogen storage. Could the authors explain the rationale for using/creating carbon fibers, their advantages and disadvantages, etc?
Line 32: hydrogen is indeed lightweight, which is a challenge more than an advantage once we consider the volumetric storage density. Please comment on this.
Line 37: in addition to liquefaction, compression should be indicated as a typical hydrogen storage technique. Liquafaction is also used, but requires much lower temperatures.
Figure 1: in the bottom right panel, Pd nanoparticles appear deposited at the surface of the fibers rather than inside the pores; could the authors comment in the text, where appropriate, whether this is the case or if Pd NPs also enter the pores?

Response 1. We used carbon fiber as hydrogen storage media because of some advantages that enables to provide excellent hydrogen storage capacity due to its high surface areas, tunable texture structure, and low gas-solid interactions. The related mention was added and rewritten in the introduction part of the revised manuscript as follows.

Page 2, line 50

~ Among these, carbon-based porous materials, especially porous carbon fibers draw attention to scientists as one of the most appropriate candidates for hydrogen storage because of their high pore volume, surface area, tunable texture structure, low gas-solid interaction, and excellent chemical, thermal stability [8,17]. Another important reason that carbon-based porous fibers are suitable as hydrogen storage materials ~

Related references on the rationale of the use of carbon fiber as materials for hydrogen storage were also added and listed as follows.

Page 13, line 432

  1. Hwang, S.-H.; Choi, W.M.; Lim, S.K. Hydrogen storage characteristics of carbon fibers derived from rice straw and paper mulberry. Mater. Lett. 2016, 167, 18–21, doi:https://doi.org/10.1016/j.matlet.2015.12.118.

On the other hand, as reviewer pointed out, the volumetric storage density of hydrogen is a very important challenge to solve. And so, in introduction part, we mentioned not only the lightweight character of hydrogen as a matter to solve but also the compression method as one of the method for hydrogen storage. In present, although the compression and liquefaction have been commonly used, the storage of hydrogen molecules via porous materials should be considered as an attractive technique due to the fast reaction kinetics, high adsorption capacity which could lead to smaller storage volume, and improved level of safety compared with that of the compressed-gas storage process.

The related mention was added and rewritten in the introduction part of the revised manuscript as follows.

Page 1, line 38

~ In general, hydrogen is stored in one of four different ways: high-pressure compression method, liquefaction within a high-pressure tank, storage in a solid-state hydride, and storage in a porous material [12]. Although the compression and liquefaction were commonly used, the storage of hydrogen molecules via porous materials has been considered as an attractive technique due to the fast reaction kinetics, high adsorption capacity, and improved level of safety compared with that of the compressed-gas storage process. Moreover, liquefaction of hydrogen has its own challenges: high energy cost and risk of explosion, requiring the extremely low temperatures.~

Regarding the Pd nanoparticle deposition, we concluded that the electro-deposited Pd nanoparticles were deposited at the surface of the fibers from the SEM results. However, it was expected that it is possible to deposit the Pd nanoparticles inside the pores near the edge of the fiber. We will conduct a further research based on your valuable advice in near future. Thank you for giving insight into further study once again.

Related discussions regarding the deposition behavior of Pd nanoparticles are in the results and discussion part as follows.

Page 10, line 327

As expected, the Pd nanoparticles were really found as aggregates on the APCF_H 0.3 and to partially cover most of the pores on the surface of the fibers as the concentration of the PdCl2 solution increases

Comment 2. Materials and Methods. In line 134 (and elsewhere in the text) the authors refer to “best porosity” which is not clear as it could refer to surface area, pore size distribution, pore volume etc. Please use a more specific description - I assume this refer to “highest surface area” in this context.

Response 2. We are sorry for making you confused with the unclear terminology. The sentence was revised as follows for clarifying the explanation.

Page 4, line 137

Among the APCFs, APCF_H 0.3 possessed the highest surface area.

Comment 3. Results and discussion. Line 245: unit is missing after 1.08; 
Regarding the method: could the authors discuss the possibility of doing alkaline activation+carbonisation in a single step (steps c and d on Figure 1)? Is it something worth trying, that has been attempted, etc?
Line 285: a reference is needed after the statement “it is reported that narrow pores ...”
Line 287: the sentence starting with “However, the heat of adsorption...” is not clear: small pores (<1 nm) are generally associated to a higher heat of adsorption (or to stronger interactions between the gas and the adsorbent) which is generally seen as a good thing for adsorption-based storage. Please clarify this sentence.
Line 316: The sentence should read “pore volume falls by 4.1%” instead of “to 4.1%” based on numbers from Table 2.
In the section about palladium deposition, the authors should comment on whether the addition of palladium is worth the cost/effort considering the relatively small increase in hydrogen storage capacity (this would also fit in the conclusion section, see my comment below).

Response 3. Firstly, according to the reviewer’s comment, the missing unit is added and revised. We are sorry for this mistake and thank for your comment.

Secondly, chemical activation process is composed of the carbonization which is conducted under inert gas atmosphere at a certain temperature and alkaline activation which includes the impregnation with chemical agents, thermal treatment, and process of washing and drying. Generally, in order to obtain the highest porous carbon structure, it was reported that the formation of carbon framework should be priorly required before alkaline activation such as etching process, gasification, and lattice expansion. Therefore, in the view point of higher porosity, it would be more efficient to conduct alkaline activation and carbonization in separate steps rather than in a single step. In fact, if the polymeric fiber is treated with alkaline activation and carbonization at once, the microporous structure would not form properly. However, it is thought to be meaningful to compare how different it is between sperate process and single step one in porosity.

Thirdly, the sentence in line 285, 287, 316 were revised as follows.

Page 9, line 294

~, it is reported that narrow pores under 1.5 nm improve the hydrogen storage because the narrow pores enhance the interaction energy between hydrogen molecules and pore walls of carbon materials [36].

Related reference is added in the manuscript.

Page 14, line 503

  1. Czakkel, O.; Nagy, B.; Dobos, G.; Fouquet, P.; Bahn, E.; László, K. Static and dynamic studies of hydrogen adsorption on nanoporous carbon gels. Int. J. Hydrogen Energy 2019, 44, 18169-18178, doi:https://doi.org/10.1016/j.ijhydene.2019.05.131

Regarding the sentence starting with “However, the heat of adsorption...”, we rewrite the manuscript to clarify the sentence according to reviewer’s comment. As reviewer pointed out, small pores (<1 nm) corresponding to micropore are closely associated to adsorption-based hydrogen storage capacity. However, optimal pore size for hydrogen storage is also known to be 0.6~0.7 nm since dynamic hydrogen molecules could be trapped properly on pores around the range of optimal pore size, but, would not easily enter the inside of pores in case of smaller pore size (< 0.5 nm). Related reference was also added in the revised manuscript.

 Page 9, line 294

~ However, the pore size below 0.5 nm is not sufficient for hydrogen storage because dynamic hydrogen molecules would not enter the inside of the pore [37]. Therefore,~

Page 14, line 503

  1. Im, J.S.; Park, S.-J.; Kim, T.J.; Kim, Y.H.; Lee, Y.-S. The study of controlling pore size on electrospun carbon nanofibers for hydrogen adsorption. Journal of Colloid and Interface Science 2008, 318, 42-49, doi:https://doi.org/10.1016/j.jcis.2007.10.024.

The sentence “pore volume falls to 4.1%” was corrected to “pore volume falls by 4.1%”

Page 10, line 323

~ the BET surface area falls to 2611 m2 g-1, the total pore volume falls to 1.42 cm3 g-1, and the percentage of micropore volume in total pore volume falls by 4.1%.

Finally, Regarding to the Pd deposition, the micropores created via carbonization and chemical activation were partially blocked by the deposition of Pd nanoparticles on the surface of APCF H 0.3, which could be lead to a low increase in hydrogen storage capacity. That is, the enhancement in hydrogen storage capacity via spillover effect is relatively low (0.3 wt%) due to deposition of Pd nanoparticles since the micropores of activated porous carbon fiber were partially covered by the deposition of Pd nanoparticles. However, it was thought to be a meaningful result in that, for the application of metal-deposited carbon fiber in hydrogen storage, the effect of metal nanoparticles on pore chracteristics in metal-deposited carbon fiber must be considered together with spillover effect.

The related discussions regarding the effect of Pd nanoparticles are added in the results and discussion part as follows.

Page 11, line 335

~ Interestingly, when the PdCl2 concentration is increased, the hydrogen storage capacity saturates. It seemed that electro-deposited Pd nanoparticles became to partailly block the micropores on the APCF_H 0.3, resulting in decreasing the fraction of micropore volume that could be a helpful factor for hydrogen storage capacity. Simultanously, the hydrogen storage capacity increases with the spillover effect of Pd nanoparticle that involves the following steps: (i) chemisorptive dissociation of hydrogen molecules on the Pd nanoparticles; (ii) diffusion of hydrogen atoms from Pd nanoparticles (hydrogen-rich) to the carbon substrate (hydrogen-poor); and (iii) storage of hydrogen atoms at pores of carbon substrate [20,21,47]. Therefore, between two conflicting effects, it is necessary to investigate which effect dominates on hydrogen storage capacity. To estimate the extent of the improvement in the hydrogen storage capacity caused by the spillover effect alone, the hydrogen storage capacity of Pd 0.5/APCF_H 0.3 via physisorption, 4.78 wt%, was calculated by the above mentioned empirical equation (Eq 1, Figure S1 and S3). The assumed improvement in hydrogen storage capacity resulting from the spillover effect was calculated by subtracting the calculated hydrogen storage capacity from the experimental result. Hence, the improvement caused by the spillover effect was determined to be around 0.67 wt%. These results indicate that there is a competitive relationship between the physisorption by alkali activation and chemisorption by the spillover effect [48-50]. Therefore, for the application of metal-deposited carbon fiber in hydrogen storage, the effect of metal nanoparticles on micropore volume and pore size of the metal deposited-carbon fiber would have to be considered together with spillover effect. [51,52].

Comment 4. Conclusions. This section is mostly a summary of the rest of the article. The authors should give here a broader perspective about their findings in the general context of porous materials for hydrogen storage. For example, where would it be advantageous to use such fiber-based materials? Is it worth adding palladium? Critically commenting the results would significantly enrich the conclusion and the article itself.

Response 4. The conclusions are revised based on the reviewer’s comments.

Page 11, line 377

~ In summary, we developed a new strategy to achieve a high-capacity means of hydrogen storage on the basis of carbon fibers by controlling the surface porosity. The core strategy is associated with forming pores both exterior and interior of the carbon fiber using the polyvinyl alcohol/polyacrylonitrile (PVA/PAN) bi-component fibers. The difference in the solubility of PVA and PAN in water is closely related to the formation of the internal pores. In this method is found to be very effective to improve the pore characteristics of carbon fibers by way of successive chemical activation. Finally, the Pd nanoparticles were electrodeposited on the surface of the carbon fiber for further improvement of the hydrogen storage capacity, however, it was also found that the enhancement in hydrogen storage capacity via spillover effect is relatively low (~0.3 wt%) due to deposition of Pd nanoparticles since the micropores of activated porous carbon fiber were partially covered by the deposition of Pd nanoparticles. For the application of metal-deposited carbon fiber in hydrogen storage, the effect of metal nanoparticles on pore characteristics in metal deposited-carbon fiber would have to be considered together with spillover effect. We believe that the proposed method which involves the control of the surface morphology of the carbon fibers and Pd deposition on their surfaces could provide high hydrogen storage capacity.

Comment 5. In general: most figures are low-resolution and sometimes difficult to interpret; can the authors supply high-resolution version of the figures for the published version of the article. 

Response 5. We apologize for any inconvenience caused by providing low-resolution images. Now, we will provide high-resolution version of the figures for the process to submission system.

Reviewer 2 Report

Manuscript Number: nanomaterials-1264052

Title: The enhanced hydrogen storage capacity of carbon fibers: The effect of hollow porous structure and surface modification

This manuscript (MS) presents synthesis of highly porous carbon fibers (HPCFs) and Pd nanoparticle decorated HPCFs for H2 storage via high pressure physisorption at 77 K. I recommend publication of this MS after minor revision.

  1. The actual effect of Pd Nanoparticles on enhancing the H2 uptake is not considerable, required to explain this property. For example, the hydrogen spillover effect with Pd and other nanoparticle based carbon materials and various different particle sizes and their effect on the H2 storage capacity at 77K and at or above room temperature was collectively discussed in the following article, please refence this in MS, “Prog. Mater. Sci. 2015, 69, 1-60. http://dx.doi.org/10.1016/j.pmatsci.2014.10.004.” Also, along this, the following closely relevant articles can enhance the relevance and understanding of the topic, materials and uptake capacity variations/relations with respect to the synthesis, porosity, as well as influence other structural parameters or experimental conditions. A wide range of porous networks with controlled surface and interface properties is demonstrated to enhance their performance – reference these relevant articles; “ Energy Mater. 2020, 10, 1903649. https://doi.org/10.1002/aenm.201903649; Science 368, 297–303 (2020), https://doi.org/10.1126/science.aaz8881; Electrochimica Acta 325 (2019) 134941, https://doi.org/10.1016/j.electacta.2019.134941; Energy Environ. Sci., 2017, 10, 2552—2562, https://doi.org/10.1039/C7EE02616A.
  2. On pages 6 and 9, Correct Table 1 and Table 2 as Table 2 and Table 3, respectively.
  3. On page 9, put the following into the experimental section and also connect eq (1) in the main text. “Hydrogen storage capacity (wt%) = 1.41 × 10-3× SSA carbon fiber (m2 g-1) (1) where SSA carbon fiber is the specific surface area of the as-prepared carbon samples.”
  4. On page 11, correct “…abovementioned…” as “…above mentioned…”

Author Response

Responses to the Reviewer’s Comments

Manuscript ID: Nanomaterials-1264052

Title: The enhanced hydrogen storage capacity of carbon fibers: The effect of hollow porous structure and surface modification

Dear Reviewers,

First of all, I’d like to express my sincere and thanks to Reviewers who read my manuscript with deep concerns and to valuable comments.

According to reviewer’s comments, the manuscript was revised and checked to avoid some typical mistakes and expression or notation to cause misunderstanding.

We appreciate kind comments and suggestion of reviewer again. Our responses to these comments are listed below, and the manuscript has been accordingly revised. All the changes were represented in blue color in revised manuscript.

Thank you for your kindness and help

Sincerely yours,

Response to the Reviewer 2’s comments

This manuscript (MS) presents synthesis of highly porous carbon fibers (HPCFs) and Pd nanoparticle decorated HPCFs for H2 storage via high pressure physisorption at 77 K. I recommend publication of this MS after minor revision.

Comment 1. The actual effect of Pd Nanoparticles on enhancing the H2 uptake is not considerable, required to explain this property. For example, the hydrogen spillover effect with Pd and other nanoparticle based carbon materials and various different particle sizes and their effect on the H2 storage capacity at 77K and at or above room temperature was collectively discussed in the following article, please refence this in MS, “Prog. Mater. Sci. 2015, 69, 1-60. http://dx.doi.org/10.1016/j.pmatsci.2014.10.004.” Also, along this, the following closely relevant articles can enhance the relevance and understanding of the topic, materials and uptake capacity variations/relations with respect to the synthesis, porosity, as well as influence other structural parameters or experimental conditions. A wide range of porous networks with controlled surface and interface properties is demonstrated to enhance their performance – reference these relevant articles; “ Energy Mater. 2020, 10, 1903649. https://doi.org/10.1002/aenm.201903649; Science 368, 297–303 (2020), https://doi.org/10.1126/science.aaz8881; Electrochimica Acta 325 (2019) 134941, https://doi.org/10.1016/j.electacta.2019.134941; Energy Environ. Sci., 2017, 10, 2552—2562, https://doi.org/10.1039/C7EE02616A.

Response 1. Thanks for providing relevant references and valuable advice to help us understand the reason that hydrogen storage capacity is increased with a small amount through spillover effect. In our case, the micropores created via carbonization and chemical activation were found to be partially blocked by the deposition of Pd nanoparticles (average size: ~ 40 nm), which resulted in a low increase in hydrogen storage capacity. According to the literature, it is reported that the enhancement of hydrogen uptake via spillover effect is significantly influenced by the size and phase of Pd nanoparticles. In particular, Pd nanoparticles with sub 10 nm were found to be more efficient in hydrogen uptake by spillover effect at room temperature. We concluded that the reason for the relatively low increase in the hydrogen storage capacity of carbon fiber by spillover effect was considered to be due to the pore-blocking of Pd nanoparticles. And therefore, for the application of metal-deposited carbon fiber in hydrogen storage, the effect of metal nanoparticles on micropore volume and pore size of the metal deposited carbon fiber would have to be considered together with spillover effect.

Page 11, line 335

~ Interestingly, when the PdCl2 concentration is increased, the hydrogen storage capacity saturates. It seemed that electro-deposited Pd nanoparticles became to partially block the micropores on the APCF_H 0.3, resulting in decreasing the fraction of micropore volume that could be a helpful factor for hydrogen storage capacity. Simultanously, the hydrogen storage capacity increases with the spillover effect of Pd nanoparticle that involves the following steps: (i) chemisorptive dissociation of hydrogen molecules on the Pd nanoparticles; (ii) diffusion of hydrogen atoms from Pd nanoparticles (hydrogen-rich) to the carbon substrate (hydrogen-poor); and (iii) storage of hydrogen atoms at pores of carbon substrate [20,21,47]. Therefore, between two conflicting effects, it is necessary to investigate which effect dominates on hydrogen storage capacity. To estimate the extent of the improvement in the hydrogen storage capacity caused by the spillover effect alone, the hydrogen storage capacity of Pd 0.5/APCF_H 0.3 via physisorption, 4.78 wt%, was calculated by the above mentioned empirical equation (Eq 1, Figure S1 and S3). The assumed improvement in hydrogen storage capacity resulting from the spillover effect was calculated by subtracting the calculated hydrogen storage capacity from the experimental result. Hence, the improvement caused by the spillover effect was determined to be around 0.67 wt%. These results indicate that there is a competitive relationship between the physisorption by alkali activation and chemisorption by the spillover effect [48-50]. Therefore, for the application of metal-deposited carbon fiber in hydrogen storage, the effect of metal nanoparticles on micropore volume and pore size of the metal deposited-carbon fiber would have to be considered together with spillover effect. [51,52].

The related reference that you provide is added in the manuscript

Page 15, Line 536

  1. Chen, Z.; Li, P.; Anderson, R.; Wang, X.; Zhang, X.; Robison, L.; Redfern, L.R.; Moribe, S.; Islamoglu, T.; Gómez-Gualdrón, D.A., et al. Balancing volumetric and gravimetric uptake in highly porous materials for clean energy. Science 2020, 368, 297-303, doi:10.1126/science.aaz8881.
  2. Blankenship, L.S.; Mokaya, R. Cigarette butt-derived carbons have ultra-high surface area and unprecedented hydrogen storage capacity. Energy Environ. Sci. 2017, 10, 2552-2562, doi:10.1039/C7EE02616A.
  3. Gadipelli, S.; Guo, Z.X. Graphene-based materials: Synthesis and gas sorption, storage and separation. Progress in Materials Science 2015, 69, 1-60, doi:https://doi.org/10.1016/j.pmatsci.2014.10.004.
  4. Chen, T.; Zhou, Y.; Luo, L.; Wu, X.; Li, Z.; Fan, M.; Zhao, W. Preparation and characterization of heteroatom self-doped activated biocarbons as hydrogen storage and supercapacitor electrode materials. Electrochimica Acta 2019, 325, 134941, doi:https://doi.org/10.1016/j.electacta.2019.134941.
  5. Gadipelli, S.; Howard, C.A.; Guo, J.; Skipper, N.T.; Zhang, H.; Shearing, P.R.; Brett, D.J.L. Superior Multifunctional Activity of Nanoporous Carbons with Widely Tunable Porosity: Enhanced Storage Capacities for Carbon-Dioxide, Hydrogen, Water, and Electric Charge. Advanced Energy Materials 2020, 10, 1903649, doi:https://doi.org/10.1002/aenm.201903649.

Comment 2. On pages 6 and 9, Correct Table 1 and Table 2 as Table 2 and Table 3, respectively.

Response 2. As the reviewer pointed out, we were trying to find a typo in the numbering of table. However, we didn’t find any mistakes in the numbering. Could you please point out again the exact meaning of the comment 2?

Comment 3. On page 9, put the following into the experimental section and also connect eq (1) in the main text. “Hydrogen storage capacity (wt%) = 1.41 × 10-3× SSA carbon fiber (m2 g-1) (1) where SSA carbon fiber is the specific surface area of the as-prepared carbon samples.”

Response 3. Thank you for the comment. We put the equation in the experimental section and related explanation was addressed in the text as follows.

Page 4, line 154

~ at 77 K (ISOSORP-HyGrA, Rubotherm, Germany). Correlation between the hydrogen storage capacity and specific area of the as-prepared carbon samples is calculated as the following equation.

Hydrogen storage capacity (wt%) = K × SSA carbon fiber (m2 g-1) (1)

where K is a slope between hydrogen storage capacity and specific surface are of the as-prepared carbon samples and SSA carbon fiber is the specific surface area of the as-prepared carbon samples. ~

Page 9, line 278

According to the equation 1, it showed that as-prepared carbon fibers can adsorb the 1.41 × 10-3 wt% hydrogen gas in every unit specific surface area at 77 K (Figure S1).

Comment 4. On page 11, correct “…abovementioned…” as “…above mentioned…”

Response 4. The sentence was revised as the reviewer points out.

Page 11, line 346

hydrogen storage capacity of Pd 0.5/APCF_H 0.3 via physisorption, 4.78 wt%, is calculated by the above mentioned empirical equation (Eq 1, Figure S1 and S3).
